# Vegetation Succession for 12 Years in a Pond Created Restoratively

**DOI:** 10.3390/biology13100820

**Published:** 2024-10-13

**Authors:** Chang-Seok Lee, Dong-Uk Kim, Bong-Soon Lim, Ji-Eun Seok, Gyung-Soon Kim

**Affiliations:** 1Department of Bio & Environmental Technology, Seoul Women’s University, Seoul 01797, Republic of Korea; bs6238@swu.ac.kr (B.-S.L.); jus826@swu.ac.kr (J.-E.S.); 2National Institute of Ecology, Seocheon 33657, Republic of Korea; kdw2782@nie.re.kr (D.-U.K.); ecokimgs@nie.re.kr (G.-S.K.)

**Keywords:** biological integrity, creation effect, ecological stability, Najeoer Pond, National Institute of Ecology

## Abstract

**Simple Summary:**

The Najeoer Pond, which is the object of this study, was created to be utilized as a test bed for the restoration of endangered species and for carrying out ecological education on the pond ecosystem. To achieve these goals, the Najeoer Pond was created for diversity and stability purposes by first securing various water depths imitating natural lagoons and inducing the establishment of various vegetation types according to the depth. As a result of analyzing the vegetation settlement process for 12 years after its creation through vegetation cover, species composition, species diversity, and the ratio of exotic species, and comparing the results with those of natural lagoons, the Najeoer Pond has been shaped to resemble the diversity and stability of natural lagoons. The carbon absorption capacity of the littoral vegetation established around the pond was significantly higher than that of forest vegetation. In this respect, the Najeoer Pond was also deemed to have a high creation effect in terms of ecological function. Therefore, a stable pond with diversity can be created because vegetation diversity can be guaranteed as a result of securing the ecological diversity of various water depths.

**Abstract:**

The Najeoer Pond was created in a rice paddy as a part of a plan to build the National Institute of Ecology. To induce the establishment of various plants, the maximum depth of the pond was 2.0 m, and diverse depths were created with a gentle slope on the pond bed. When introducing vegetation, littoral and emergent vegetation were first introduced to stabilize the space secured for the creation of the pond, whereas the introduction of other vegetation was allowed to develop naturally. In this pond, floating, emergent, wetland, and littoral plants have been established to various degrees, reflecting the water depth and water table. As a result of stand ordination, based on vegetation data obtained from the created Najeoer Pond and a natural lagoon selected as the reference site, the species’ composition resembled that of the reference site. Diversity, based on vegetation type, community, and species, tended to be higher than that of the reference site. The proportion of exotic species increased due to the disturbance that occurred during the pond creation process but continued to decrease as the vegetation introduced during the creation of the pond became established. Considering these results comprehensively, the restorative treatment served to increase both the biological integrity and ecological stability of the pond and, thus, achieved the creation goal from the viewpoint of the pond structure.

## 1. Introduction

Fast-paced climate and land-use changes are negatively affecting the structure and function of ecosystems [1,2,3,4]. As the quality degradation of ecosystems around the world progresses rapidly, initiatives to mitigate these devastating impacts are increasing [5,6,7] and the topic of ecological restoration is thus receiving ever greater attention worldwide [8,9,10].

As a mountainous country where mountainous areas account for about 65% of the country, Korea does not have a large wetland area. With a continental climate (Asian monsoon climate), most of the rainfall is concentrated in the rainy season, making it difficult to maintain wetlands. In addition, rice, a staple food in Korea, is an aquatic plant; therefore, many wetlands have been converted into rice paddies. However, in recent years, many of these rice fields have been transformed into urban areas [11,12]. Consequently, the wetlands have been severely damaged and they are less able to perform their traditional functions. However, due to recent social changes, caused by economic growth, the level of environmental awareness has increased significantly in these regions, and the restoration of damaged wetlands, including ponds, is actively progressing.

However, except for a few advanced countries in this field, the ecological restoration currently underway simply refines the esthetics rather than performing actual restoration, which is urgently required. It is believed that this low level of ecological restoration is due to insufficient understanding of the true meaning of ecological restoration, as well as negligence in terms of evaluating the effectiveness of restoration [5,13,14,15,16].

Ecological restoration aims to restore intact and healthy natural conditions before they are destroyed irreversibly. In other words, environmental restoration is an ecological technology that heals nature, which has been negatively impacted, by imitating the structure and function of intact nature, and through this, it aims to provide habitats for various wildlife and a future environment for humankind [5,16,17,18,19]. Ecological restoration is considered to improve the productivity of degraded lands, conserve biodiversity, and supplement or reinforce lost or damaged ecosystems [5,6,16,19,20,21,22]. Human aid is sometimes needed to restore damaged ecosystems and prevent further damage, and habitat improvement and the introduction of species may be required to allow the process of recovery to begin [16,19,23,24,25].

Ecological restoration, as a process to help the recovery of ecosystems that have been degraded, damaged, or destroyed, progresses through a series of processes as follows: diagnostic assessment, collection of reference information, preparation of a restoration plan, based on the result of the diagnostic assessment and reference information, restoration implementation, monitoring of the establishment process and adaptive management based on the result, and evaluation of the restoration effects [5,16,19].

Numerous restoration projects are carried out around the world every year; however, the success of restoration is not well known, due to a lack of evaluation of the projects’ effectiveness and limited information regarding their success [13,14,15,26,27]. Without a comprehensive evaluation of a project, the project cannot develop because there is no information about how its achievements differ depending on its approach. Standardized monitoring, evidence-based evaluation, and information delivery are invaluable in terms of planning future projects. Investment in the evaluation of restoration effects will also contribute to increasing the relevance and applicability of ecological research for restoration [8,28,29,30,31]. In this respect, an evaluation of the restoration effect is essential for the development of ecological restoration.

The pond is an important habitat for conserving biodiversity and plays a key role in providing humans with diverse benefits such as climate change mitigation and adaptation, habitat creation and maintenance of biodiversity conservation, water purification, flood mitigation, and cultural benefits [32]. Water-related ecosystems, such as rivers, lakes, and ponds, etc. are ecosystem complex that aquatic zone and littoral zone are combined [33]. The aquatic zone is divided into several zones according to the growth form of the established plants: emergent plants rooted at the bottom, but with other vegetative organs such as leaves and stems and reproductive organs such as flowers in the air; floating-leaved plants rooted at the bottom, but with floating leaves; submerged plants with all organs below the water surface except for the reproductive organs; and free-floating macrophytes with all plant organs including the roots on the water surface. These macrophytes of various growth forms generally occur in different zones of the aquatic area, usually depending on the water depth [34,35,36]. The littoral zone is the land next to the water bodies of ponds. Littoral ecosystems play crucial roles in connecting upland forests and aquatic habitats and have unique functions. The littoral ecosystems surround the pond and protect and promote water quality, aquatic ecosystem health, and shore stability [37,38,39].

However, the littoral zones are often degraded by a variety of human disturbances [33,40,41]. Moreover, the aquatic zone is not equipped with vegetation zones of different growth forms, as it does not have various water depths [42,43]. In this regard, the restoration of the pond to a complete system and the establishment of diversity is required.

This study aims to highlight restoration effects by analyzing the process of establishing a pond created within the campus of the National Institute of Ecology in Korea. The establishment of restorative treatment was analyzed based on changes in vegetation maps, growth forms and species composition, diversity of vegetation types, communities and species, and the percentage of exotic species. The restoration effects were evaluated by comparing the information with that of the reference sites.

We hypothesized that the pond in our study would resemble a natural lagoonscape over time after creation. Evaluating the vegetation changes in a created pond and the carbon absorption capacity over time provides a novel understanding of pond creation evolution and serves as a possible indicator parameter for evaluating the progress or success of pond creation projects. It is also useful to understand how long it could take after pond creation for the pond ecosystem to be comparable to a natural pond, which is helpful for managers responsible for approving, funding, and designing pond creation projects as well as achieving multiple ecosystem service outcomes.

## 2. Materials and Methods

### 2.1. Study Area

The construction of the National Ecological Institute (abbreviated as NIE hereafter) began as an alternative project not only to conserve the tidal flat but also to develop the region, instead of constructing the Janghang Industrial Complex, which was planned to reclaim the Seocheon tidal flat. To achieve this goal, all facilities in the NIE campus were prepared by thoroughly applying the principle of ecology, particularly restoration ecology. The basis of ecosystem management in the NIE was to induce passive restoration [44,45].

The place where the Najeoer Pond is located was originally rice paddies (Figure 1). Topographically, this was the bottom of the basin where the NIE research complex was located, where a relatively large amount of water could be collected (Figure 1).

Cheonjin Lagoon is located in Goseong-gun, Gangwon province, central-eastern Korea (Figure 1A). The maximum depth of Cheonjin Lagoon is about 2 m, and from there, it slopes gently toward the land, resulting in various depths and thus various aquatic plants. Cheonjin Lagoon is a place where endangered plants such as *Nymphaea tetragona* var. *minima* and *Brasenia schreberi* are distributed, as well as various aquatic plants and littoral vegetation, established depending on the water depth. Therefore, it is known as the lagoon with the best preservation conditions among the 19 natural lagoons distributed on the east coast of South Korea [46]. In consideration of these ecological conditions, Cheonjin Lagoon was selected as a reference site for this study.

### 2.2. Creation Practice

The Najeoer Pond was created in the paddy fields. Initially, a pool was formed, the deepest point of which was 2 m, which sloped gently in the paddy field of the flatland. At this time, the topsoil, with fine particles of paddy soil, was stored for future use. After the pool was dug, the bottom was covered with the topsoil of the rice paddy, which was stored to minimize the outflow of water through the floor. Afterwards, various water depths were prepared so that free-floating, floating-leaved, submerged, emergent, wetland, and littoral plants could be evenly established. The introduction of vegetation aimed to emulate the typical vegetation structure of a pond equipped with plant zones with all the aforementioned growth forms. Littoral and emergent vegetation was first introduced to stabilize the space required for the creation of the pond, and the introduction of other vegetation was left to develop naturally.

The reference information relating to the creation of the pond was collected from analyses of the Ramsar registered wetlands, the backwater wetlands of the river, and the successional process of the abandoned rice paddy.

### 2.3. Vegetation Survey

The aerial images, taken by a drone, were used to identify vegetation types and landscape boundaries. These vegetation types and landscape elements were confirmed by field checks. The landscape attributes were overlapped onto topographical maps at a scale of 1:5000. Patches smaller than 1 mm on the map were excluded from this study because of the uncertainty of their sizes and shapes [47]. Mapping was performed using the ArcView GIS (Geographic Information System), and landscape ecological analyses were conducted using the ArcView GIS software (Version 10.1) [48].

A vegetation survey was conducted before, during, and on the 4th and 12th years after the creation of the pond. Vegetation samples from the created pond and the reference pond were compared using several metrics, including species composition, diversity, and the percentage of exotic species. The vegetation survey was conducted during the summer (i.e., June to August) in all survey years by recording the cover classes of plant species in quadrates that were randomly established [49]. The Braun-Blanquet [50] scale was used to measure plant cover. The ordinal cover of the Braun-Blanquet scale was converted to the median value of the percentage cover range for each cover class and then subjected to a Detrended Correspondence Analysis [51]. For the vegetation survey, quadrates sized 1, 4, and 100 m^2^ were used for grassland, shrubland, and tree-dominated areas, respectively. Diversity was compared based on the number of vegetation types, communities, and species in the created pond. Species diversity was compared using species rank–abundance curves, which graphically depict patterns of species diversity and dominance [52,53]. The percentage of exotic species was calculated by dividing the number of exotic species by the total number of species [11].

## 3. Results

### 3.1. Changes in Vegetation Cover

No vegetation types could be expressed as a vegetation map in the rice fields before the creation of the Najeoer Pond. The creation of this pond, which was implemented in 2011, was carried out by introducing the *Z. latifolia* community, the *P. australis* community, and the *S. pierotii* community. During the 4th year after its creation, in addition to the three vegetation types introduced, the *T. japonica* community and the *Leersia japonica* community established themselves naturally. During the 12th year after the creation of the pond, the *T. orientalis* community and the *M. sacchariflorus* community were established naturally (Figure 2).

### 3.2. Changes in Species Composition

As a result of stand ordination, based on vegetation data collected from the rice fields before the creation of the Najeoer Pond, the arrangement of the stands was distributed in the following order: *Persicaria thunbergii* community, *Echinochloa crusgalli* var. *echinatum* community, and *Alopecurus aequalis* community from left to right on Axis I. During its year of creation, the *Z. latifolia* community, the *S. pierotii* community—*Panicum bisulcatum* community—*Kummerowia striata* community—*B. frondosa* community, and the *P. australis* community were arranged in the aforementioned order from left to right on Axis I. During the 4th year, the vegetation types were arranged in the following order: *S. pierotii* community—*K. striata* community, *P. australis* community—*Amphicarpaea bracteata* subsp. *edgeworthii* community, *Miscanthus sacchariflorus* community, and *Z. latifolia* community—*Leersia japonica* community—*T. japonica* community from left to right on Axis I. During the 12th year, the *S. pierotii* community—*M. sacchariflorus* community, *Leersia sayanuka* community—*A. bracteata* subsp. *edgeworthii* community*—H. japonicus* community—*Calamagrostis epigeios* community—*P. australis* community, *Z. latifolia* community—*Juncus effusus* var*. decipiens* community, *T. orientalis* community—*L. japonica* community, and the *T*. *japonica* community were distributed in the aforementioned order from left to right on Axis I. As shown above, in the results of stand ordination, based on the vegetation data, the arrangement of the stands tended to reflect the water depth at which each stand was located without any relation to the survey year (Figure 3).

On the other hand, compared to the growth form of the dominant species making up the plant community that emerged, floating-leaved, emergent, wetland, and littoral plants appeared in the Najeoer Pond; however, the plant community dominated by wetland plants did not appear in Cheonjin Lagoon (Figure 3).

As a result of stand ordination based on vegetation data collected from the natural Cheonjin Lagoon, the arrangement of the stands was distributed in the order of the *Nymphaea tetragona* var. *minima* community, the *Brasenia schreberi* community, the *Potamogeton wrightii* community—*Trapa japonica* community, the *Nymphoides peltata* community—*Zizania latifolia* community, and the *Phragmites australis* community—*Salix pierotii* community—*S. integra* community, as shown from left to right on Axis I (Figure 4). Plant communities, formed by floating-leaved plants, such as the *Nymphaea tetragona* var. *minima* community, the *Potamogeton wrightii* community, the *Brasenia schreberi* community, the *Trapa japonica* community, and the *Nymphoides peltata* community, tended to be arranged in the aforementioned order from the lower to the upper sections on Axis II (Figure 4).

As a result of stand ordination, based on vegetation data collected from both the created Najeoer Pond and the natural Cheonjin Lagoon, the species composition of vegetation established on both sites was found to be very similar except for the floating-leaved plant zone (Figure 4).

The results of stand ordination were expressed by classifying the growth form of the dominant species (Figure 5). The results show that in both the created Najeoer Pond and the natural Cheonjin Lagoon, communities dominated by floating-leaved plants, emergent plants, wetland plants, and littoral plants have become established, and their spatial distribution depends on the water depth.

### 3.3. Changes in Species Diversity

The number of vegetation types, plant communities, and plant species in the created Nanjeoer Pond increased from 0 to 7, 0 to 12, and 36 to 96, respectively (Figure 6). The numbers of communities and species that appeared in the Cheonjin Lagoon were 9 and 76, respectively, and the community and species diversity of the created Najeoer Pond was greater than that of the natural Cheonjin Lagoon (Figure 6).

As a result of analyzing the change in species diversity by the species rank–dominance curve, the species diversity showed an increasing trend over time after the creation of the pond, as shown in the aforementioned results. When comparing the species diversity of communities, this diversity was higher in the littoral plant communities compared to the aquatic plant communities, and the species diversity tended to be lower as the plant communities were established at a greater depth in both the created Najeoer Pond and the natural Cheonjin Lagoon (Figure 7).

### 3.4. Percentage of Exotic Plant Species

In the early stage of the creation of the pond, the ratio of exotic species increased significantly, due to the disturbance that occurred during the creation process. However, during the years after creation, this ratio decreased as the vegetation introduced became established (Figure 8).

## 4. Discussion

### 4.1. Spatial Distribution of Vegetation in the Created Najeoer Pond

The spatial structure of the pond as a landscape is divided into the aquatic zone and the littoral zone. The aquatic zone can be classified into the emergent, floating-leaved, submerged, and free-floating macrophyte zones, depending on their manner of growth. These various types of macroplants usually occur in different zones of the aquatic area depending on the water depth, with the emergent plants located closest to the shore, followed by the floating-leaved macrophytes and the submerged plants. Free-floating macrophytes can occur anywhere on the water surface of the system [34,35,36].

The terrestrial plant zone is divided into the wetland and littoral plant zones. The wetland plant zone is usually distinguished by the presence of shrubs and other vegetation that does not tolerate wet conditions. The littoral zone is the land next to the water bodies of ponds. Littoral ecosystems play various and unique roles by connecting upland forests and aquatic habitats. The littoral ecosystems that surround the pond protect the aquatic ecosystem and contribute to the health and stability of the system [37,38,39]. The pond shoreline and littoral zone are interconnected and work together to maintain clean water and a healthy, functioning ecosystem [34,35,36].

The vegetation maps of the Najeoer Pond during its year of creation show that the *Z. latifolia* community and the *P. australis* community are the emergent plants, and the *S. pierotii* community became established as the littoral plant (Figure 2). Since then, the *T. japonica* community, the *L. japonica* community, the *T. orientalis* community, and the *M. sacchariflorus* community have been added. In addition, wetland plants such as the Calamagrostis epigeios community and the Juncus effusus var. decipiens community have appeared, as well as littoral plants such as the H. japonicus community and the *Amphicarpaea bracteata* subsp. *edgeworthii* community, and floating plants such as *S. polyrhiza* (Figure 3), but they are not shown on the vegetation map because they did not form a sufficiently large community.

Based on these results, it was determined that the created Najeoer Pond comprised a complete spatial structure except for the submerged plant zone [34,35,36]. The eco-diversity that the plants of these various growth forms established depending on the water depth and the water table can contribute to the biodiversity of the created Najeoer Pond [36,45,54]. In fact, the numbers of communities and species investigated in the Najeoer Pond were higher than those of the Cheonjin Lagoon, which was selected as a reference site (Figure 5). The diversity of the plant community in ponds is maximized when emergent and submerged plants cover 50–80% of the pond, and more than 90% of the littoral zone is covered by vegetation. The buffer zone provides essential reproduction, wintering, and shelter for wildlife [35,36,55,56].

In addition, a healthy pond is usually covered with macrophytes appearing from the open waters through the shoreline to the littoral zone, such as submerged, free-floating, leaf-floating, emergent, wetland, and littoral plants. On the other hand, if it is degraded by eutrophication, the coverage of macrophytes decreases, whereas the coverage of phytoplankton significantly increases. In reality, the introduction of macrophytes purifies water, regulates the biological structure, and controls eutrophication in a lake [57]. Considering these results, the complete structure of aquatic ecosystems could be recommended as a means of managing water quality in countries where water quality problems are relatively severe due to the use of land for cultivating rice.

### 4.2. Creation Effects Based on Species Composition and Species Diversity

The process of restoration can be examined in terms of ecosystem structure and function [58,59], as both are heavily affected by devastation. The fundamental goal of restoration is to return damaged habitats or ecosystems to conditions close to those before they were damaged. While a complete restoration means a return to that state, there are also other levels of restoration, such as partial recovery or replacement by other systems [16,17,20,60]. In order to effectively restore degraded ecosystems or protect existing high-quality areas, it is necessary to set the properties of normal and healthy ecosystems as a reference [16,19,61]. One way of setting criteria to assess the success or failure of restoration is to define the normal biological integrity of a system and then measure its deviation from it. Integrity refers to an intact state, or a complete or unfragmented quality or condition. Biological integrity is defined as the ability to support and maintain a balanced, intact, adaptable biological system with all the elements and processes expected in the natural habitat of an area [62,63].

To evaluate the created Najeoer Pond, the ecological attributes of the pond need to be compared with those from an undisturbed reference [64,65,66]. In the present study, we compared the species composition, biodiversity, and exotic species ratio of the created Najeoer Pond with those of the natural Cheonjin Lagoon. The species composition of the created Najeoer Pond resembled the Cheonjin Lagoon (Figure 4) and diversity was higher (Figure 5), while the exotic species rate was lower compared to that of the Cheonjin Lagoon (Figure 8). Based on these results, the creation of the Najeoer Pond contributed to enhancing both biological integrity and ecological stability, and, consequently, it could be determined that this project made strides towards achieving its goal.

Biodiversity significantly increased in Najeoer Pond after its creation (Figure 3). The importance of biodiversity is based on a variety of values, including various ecological functions that bring stability to the environment [67]. Biodiversity reflects the heterogeneity of a habitat or the eco-diversity [54,68,69,70]. High biodiversity is also an indicator of the integrity of an environment, demonstrating that it is healthy and has all the necessary components [71]. The increased biodiversity of the created Najeoer Pond is due to the various water depths secured through restorative treatment. In ponds, water depth is one of the most important factors affecting vegetation establishment and community assembly [72,73,74]. A variety of water depths has been proposed to increase species diversity in created ponds [73,74,75]. In addition, Kim et al. [76], An et al. [77], and Choi et al. [78] clarified that vegetation diversity is closely related to the heterogeneity of water depth.

### 4.3. Evaluation of Effectiveness Based on the Exotic Species Ratio

In the early stages of the creation of the pond, the number of exotic species increased significantly due to the disturbance that occurred during the creation practice. Over the years after creation, this number decreased as the vegetation introduced for creation became established (Figure 6). Exotic species often extend their habitat range beyond their early settlements by leveraging favorable life history strategies [79,80,81]. The disturbed sites are usually known as the sites that exotic species with opportunistic or ruderal life history strategies favor [79,80,81], whereas a natural ecosystem with a complete system is resistant to the invasion of exotic species [81]. From these facts, we can confirm that ecological restoration heals disturbed areas and returns them to a stable state. In practice, restoration efforts often aim to restore the biodiversity and function of ecosystems lost due to disturbances [16,82]. In this context, ecological restoration that restores the intact nature of ponds by rectifying the damage could be a preventive measure against the invasion and expansion of exotic species [83,84,85,86,87]. The results of this study also demonstrate the principle of invasion ecology, namely that disturbances induce the invasion of exotic species and that intact natural ecosystems are resistant to the invasion of exotic species [79,81,88].

## 5. Conclusions

As a result of analyzing the establishment process of the created Najeoer Pond from before its creation to the 12th year after creation, its species composition began to resemble that of the natural reference lagoon. In addition, diversity in terms of vegetation type and the number of plant communities and species increased continuously over the years after the pond’s creation, reaching a level similar to that of the natural lagoon. The various water depths could be considered as factors increasing biodiversity. The percentage of the exotic species significantly increased, due to the disturbance that occurred during the creation process. However, this decreased as the vegetation introduced for pond creation was established, and maintained a lower level than that of the natural lake. This result is due to the securing of biological integrity and ecological stability through ecological restoration. Consequently, restorative treatment serves to increase both biological integrity and ecological stability and meets the restoration goal. As a result, it can be determined that restoration as a process is currently in progress in the created Najeoer Pond.

## Data Availability

The data presented in this study are available on request from the corresponding author.

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
