# Peer review of "Vegetation Succession for 12 Years in a Pond Created Restoratively"

_biology, 2024, doi:10.3390/biology13100820_

Round 1

Reviewer 1 Report (Previous Reviewer 1)

Comments and Suggestions for Authors

much improved manuscript, but I still feel that inclusion of forest structure, soil respiration and MPP MUST be removed as they are not relevant here and are not really discussed.  If they remove these I shall approve the manuscript

Comments on the Quality of English Language

need to conduct minor changes for clarity in English

Author Response

Response to reviewer’s comments (3rd)

Dear Reviewer:

Thank you for reviewers’ valuable advice and comments. We respect the reviewer's valuable advice and suggestions. So, we answered faithfully to reviewers’ advice and suggestions and revised our manuscript by reflecting reviewers’ valuable advice and comments. We

Thank you again for reviewers’ kind advice and comments.

Sincerely Yours,

Chang Seok Lee

Reviewer #1

Comments and Suggestions for Authors

much improved manuscript, but I still feel that inclusion of forest structure, soil respiration and MPP MUST be removed as they are not relevant here and are not really discussed. If they remove these I shall approve the manuscript

☞ We removed them by reflecting reviewer’s opinion. Lines 114, 195 – 257,  356 – 366, 465 – 193, 508 – 511, 728 – 769.

Reviewer 2 Report (Previous Reviewer 2)

Comments and Suggestions for Authors

I am very happy to appreciate the positive improvements in the manuscript “Vegetation succession for 12 years in a pond created restoratively”. Even the title itself sounds much better now.

I would like to emphasize that this work is extremely important for understanding the ecology of the processes of environmental restoration after human exploitation of the environment. In addition, the duration of the observations and the result deserve respect. The authors hypothesized, the pond will resemble a natural lake landscape over time. Photo 1 is a clear proof of this.

Of the flaws, I only noticed that

- on L.118 a space is lost after the word lagoon;

- the caption for Figure 7 slightly overlaps the figure.

I believe that once these minor technical flaws are corrected, the manuscript will be publishable in Biology.  

Author Response

Response to reviewer’s comments (3rd)

Dear Reviewer:

Thank you for reviewers’ valuable advice and comments. We respect the reviewer's valuable advice and suggestions. So, we answered faithfully to reviewers’ advice and suggestions and revised our manuscript by reflecting reviewers’ valuable advice and comments. We

Thank you again for reviewers’ kind advice and comments.

Sincerely Yours,

Chang Seok Lee

Reviewer #2

Comments and Suggestions for Authors

I am very happy to appreciate the positive improvements in the manuscript “Vegetation succession for 12 years in a pond created restoratively”. Even the title itself sounds much better now.

I would like to emphasize that this work is extremely important for understanding the ecology of the processes of environmental restoration after human exploitation of the environment. In addition, the duration of the observations and the result deserve respect. The authors hypothesized, the pond will resemble a natural lake landscape over time. Photo 1 is a clear proof of this.

Of the flaws, I only noticed that

- on L.118 a space is lost after the word lagoon;

☞ We changed ‘lagoonscape’ to ‘lagoon scape’. Line 115.

- the caption for Figure 7 slightly overlaps the figure.

☞ It was revised so that it does not overlap. Line 349.

I believe that once these minor technical flaws are corrected, the manuscript will be publishable in Biology.

☞ Thank you for your kind consideration.

This manuscript is a resubmission of an earlier submission. The following is a list of the peer review reports and author responses from that submission.

Round 1

Reviewer 1 Report

Comments and Suggestions for Authors

Comments on the Quality of English Language

Moderate editing of English language required.

Author Response

Response to reviewer’s comments

Dear Reviewer:

Thank you for reviewers’ valuable advice and comments. We answered faithfully to reviewers’ questions and revised our manuscript by reflecting reviewers’ valuable advice and comments.

Thank you again for reviewers’ kind advice and comments.

Sincerely Yours,

Chang Seok Lee

Reviewer #1

The authors tried to include too many topics without linking them together. The most interesting part of this study was glossed over and relates to the twelve year succession of vegetation in a created pond. If they take this approach, then I think they can have a good manuscript worthy of publication. The way it is now, it is a jumbled mess. Throughout they use term incorrectly and this must be corrected.

☞ In this study, we created a pond in a rice field that was previously a natural wetland. First of all, a puddle where the deepest point was 2 m and formed a gentle slope as move away from it, was created in the paddy field of the flatland. At this time, the topsoil with fine particles in the paddy soil was stored for future use. After the puddle was dug up, the bottom was covered with the topsoil of the rice paddy, which was stored, to minimize the outflow of water through the floor. After then, various water depths were prepared so that free floating, floating-leaved, submerged, emergent, wetland, and riparian plants could be evenly established. The introduction of vegetation aimed to have a typical vegetation structure of a pond equipped with plant zones of all growth forms mentioned above. In the restoration of vegetation, riparian and emergent vegetation were first introduced to stabilize the space secured for the restoration of the pond, and the restoration of the other vegetation was left in the natural process. Therefore, we interpreted this project as a restoration project, a creative restoration.

   In fact, restoration and succession are linked in many ways. When living things settle on newly formed or exposed land surface, the environmental conditions of the land change. Other creatures settle in such a changed environment. Succession refers to the process of changing each other as the environment and living things interact in this way. There are two types of succession: primary succession and secondary succession. The primary succession takes place when living things for the first time colonizes newly formed or exposed bedrock. Certain hardy plants and lichens with few soil requirements colonize the area first. Therefore, the primary succession proceeds with the soil formation process. Secondary succession is proceeded when the area previously occupied by living things is damaged by disturbance and then other organisms settled again. Therefore, there is a large difference in the speed of progress between the primary and secondary transitions. In this respect, restoration could be seen as a form of ‘tertiary succession’, which is a new concept that describes the human-induced changes in vegetation after primary or secondary succession.

After creation of pond, the process of establishing vegetation there was analyzed. The process was analyzed based on the vegetation cover, species composition, species diversity, and the ratio of exotic species, and the restoration effect was evaluated by comparing the results with those of the reference site. Furthermore, the carbon absorption capacity of the established vegetation was evaluated based on the net ecosystem production, and it was used as another restoration effect.

My specific comments are:

Line 2. The title of the manuscript is horrible. Readers will have no idea what Najeoer is in reference to. This is not a restoration project but a created pond succession.

☞ Succession is an ecological change that occurs naturally. However, I would like to interpret this project that promoted the progress of succession with human aid as restoration.

Line 11. The pond was created, not restored

☞ As I mentioned before, rice fields were created by transforming natural wetlands. Wetlands such as ponds are established in the lower part of the watershed like this study site. Therefore, creating a pond in an artificially converted rice field could be interpreted as an ecological restoration that heals damaged nature by imitating the system of nature.

Line 14. Riparian zone is used for stream/river banks not ponds. Perhaps littoral would be better

☞ The term "riparian zone" refers to areas around water bodies. The riparian zone refers to the broader region along the shore where the ecosystem is influenced by proximity to the lake but doesn’t specifically focus on aquatic plant growth. Therefore, the term can be used more broadly for the area around lakes or wetlands as well.

However, if I have to revise it to littoral, I will revise it.

Line 25-27. NEP not defined. Why make comparisons to forests? When I checked the net, it was as I suspected….there are not natural lakes in Korea, just a crater lake. The real story of this manuscript is the 12 year succession of vegetation in a created pond. This should be the title of the manuscript

☞ In 2.6 Calculation of carbon absorption capacity section, we defined NEP (Net Ecosystem Production. When we consider carbon absorption sources, we generally think of forests as absorption sources, so we compared carbon absorption capacity with them. We have 19 lagoons in South Korea and we regarded them as lakes.

Line 57. Actions not aids would be correct

☞ Restoration is to accelerate succession through human aid. In this study, we introduced riparian vegetation that takes a long time to establish and introduced emergent plants to ensure the stability of the pond shore.

Line 77-91. This material is not needed as it is basic background to anyone reading this manuscript. Again, riparian is not correct and should be littoral.

☞ This part refers to the need for pond restoration to provide various ecosystem services, including climate change mitigation, the complete structure of the pond, and the conditions required to maintain the ecological quality of the pond. It is judged to be essential for evaluating the restoration and restoration effects of the pond.

In addition, as was mentioned above, the term ‘riparian’ could be used more broadly for the area around lakes or wetlands as well. However, if I have to revise it to littoral, I will revise it.

Line 128. This is a new pond

☞ Yes, you are right.

Line 152. No value in talking about the vegetation before the pond was created….it was a rice paddy

☞ It is judged to be necessary data to show that this place was an artificial, but a type of wetland before restoration.

Line 175. Terrestrial trees and soil respiration are not needed at all

☞ It is necessary to evaluate the productivity of willow trees established as riparian vegetation. And measurement of soil respiration is necessary to calculate the true carbon absorption capacity of an ecosystem, NEP (Net Ecosystem Production).

Line 291. DCA Figure 5 is interesting but not really discussed to any extent…..must be redone

☞ We explained the results from Line 331 to Line 344.

Line 304. Natural lake? I disagree. This is not a proper reference site. No details were give

☞ Once severely damaged, it was restored to its natural state through ecological restoration. Therefore, many experts now recognize it as a natural lake.

I added a description of the place and explained the condition as the reference site by reflecting reviewer’s comment. Lines 175-183.

Reviewer 2 Report

Comments and Suggestions for Authors

The manuscript "The restoration effects confirmed in a Najeoer pond restored in the National Institute of Ecology, Central Western Korea" is an extremely interesting experimental field-based multi-annual study. An artificial pond was created for observation. Over a period of 12 years, the pond was surveyed for the growth of different types of plants and its biodiversity was compared with that of a natural lake. As a result of the study of the establishment process of the restored Najoer pond from pre-restoration and on the 12th year after restoration, the species composition began to resemble that of the natural reference lake. 

This conclusion, which cannot be reached without long-term routine observations, represents an important contribution to modern ecology. In general, this topic is very relevant, as aquatic ecosystems are currently under tremendous influence, such as climate change, increasing agriculture and aquaculture. 

I will only provide a few comments for the authors, which I hope will improve the manuscript.

The paragraph beginning at the L.66 should be rewritten more clearly and concisely. 

Figure 1 should be placed in the Materials and Methods section after its first mention in the text. The first part of the figure should be labelled A and the second part B. Also refer to them in the text: Figure 1A and 1B.

In my opinion, the Materials and Methods should briefly describe Cheonjin Lake with which the comparison is made: depth, location, etc.

What are the units of increasing rate of CO2 concentration in formula 1?

In the Results, it would be good to reflect the total number of species in the pond that was in the 12th year after restoration, as well as in the comparison lake. It would also be good to provide the number of common species for the pond and the lake.

Author Response

Response to reviewer’s comments

Dear Reviewer:

Thank you for reviewers’ valuable advice and comments. We answered faithfully to reviewers’ questions and revised our manuscript by reflecting reviewers’ valuable advice and comments.

Thank you again for reviewers’ kind advice and comments.

Sincerely Yours,

Chang Seok Lee

Reviewer #2

Comments and Suggestions for Authors

The manuscript "The restoration effects confirmed in a Najeoer pond restored in the National Institute of Ecology, Central Western Korea" is an extremely interesting experimental field-based multi-annual study. An artificial pond was created for observation. Over a period of 12 years, the pond was surveyed for the growth of different types of plants and its biodiversity was compared with that of a natural lake. As a result of the study of the establishment process of the restored Najoer pond from pre-restoration and on the 12th year after restoration, the species composition began to resemble that of the natural reference lake.

This conclusion, which cannot be reached without long-term routine observations, represents an important contribution to modern ecology. In general, this topic is very relevant, as aquatic ecosystems are currently under tremendous influence, such as climate change, increasing agriculture and aquaculture.

I will only provide a few comments for the authors, which I hope will improve the manuscript.

The paragraph beginning at the L.66 should be rewritten more clearly and concisely.

☞ I revised this part by reflecting reviewer’s valuable comment. Lines 87 – 105.

Figure 1 should be placed in the Materials and Methods section after its first mention in the text. The first part of the figure should be labelled A and the second part B. Also refer to them in the text: Figure 1A and 1B.

☞ I revised this part by reflecting reviewer’s valuable comment.

In my opinion, the Materials and Methods should briefly describe Cheonjin Lake with which the comparison is made: depth, location, etc.

☞ I added description for Cheonjin lake by reflecting reviewer’s valuable comment.

Lines 175-183.

What are the units of increasing rate of CO2 concentration in formula 1?

☞ On the measuring device, it is indicated as g CO2 m-2 hr-1. When calculating the amount of carbon dioxide adsorption, it is converted into ton CO-2 ha -1 yr -1 and used.

In the Results, it would be good to reflect the total number of species in the pond that was in the 12th year after restoration, as well as in the comparison lake. It would also be good to provide the number of common species for the pond and the lake.

☞ We showed them in Figure 6. The number of Common species was not shown separately because it could be read as the dominant species of the community in the ordination results.

Round 2

Reviewer 1 Report

Comments and Suggestions for Authors

see attached

Comments on the Quality of English Language

none
